# Carotid Artery Bypass Surgery of In-Body Tissue Architecture-Induced Small-Diameter Biotube in a Goat Model: A Pilot Study

**DOI:** 10.3390/bioengineering11030203

**Published:** 2024-02-21

**Authors:** Tadashi Umeno, Kazuki Mori, Ryosuke Iwai, Takayuki Kawashima, Takashi Shuto, Yumiko Nakashima, Tsutomu Tajikawa, Yasuhide Nakayama, Shinji Miyamoto

**Affiliations:** 1Department of Cardiovascular Surgery, Oita University Hospital, Oita 879-5593, Japan; kazumori@oita-u.ac.jp (K.M.); t-kawashima@oita-u.ac.jp (T.K.); shutot@oita-u.ac.jp (T.S.); nakashima-y@oita-u.ac.jp (Y.N.); smiyamot@oita-u.ac.jp (S.M.); 2Institute of Frontier Science and Technology, Okayama University of Science, Okayama 700-0005, Japan; iwai@ifst.ous.ac.jp; 3Department of Mechanical Engineering, Faculty of Engineering Science, Kansai University, Osaka 564-8680, Japan; tajikawa@kansai-u.ac.jp; 4Osaka Laboratory, Biotube Co., Ltd., Osaka 565-0842, Japan; y.nakayama@biotube.co.jp

**Keywords:** Biotube, carotid arteries, tissue engineering, vascular grafting, in-body tissue architecture

## Abstract

Biotubes are autologous tubular tissues developed within a patient’s body through in-body tissue architecture, and they demonstrate high potential for early clinical application as a vascular replacement. In this pilot study, we used large animals to perform implantation experiments in preparation for preclinical testing of Biotube. The biological response after Biotube implantation was histologically evaluated. The designed Biotubes (length: 50 cm, internal diameter: 4 mm, and wall thickness: 0.85 mm) were obtained by embedding molds on the backs of six goats for a predetermined period (1–5 months). The same goats underwent bypass surgery on the carotid arteries using Biotubes (average length: 12 cm). After implantation, echocardiography was used to periodically monitor patency and blood flow velocity. The maximum observation period was 6 months, and tissue analysis was conducted after graft removal, including the anastomosis. All molds generated Biotubes that exceeded the tensile strength of normal goat carotid arteries, and eight randomly selected Biotubes were implanted. Thrombotic occlusion occurred immediately postoperatively (1 tube) if anticoagulation was insufficient, and two tubes, with insufficient Biotube strength (<5 N), were ruptured within a week. Five tubes maintained patency for >2 months without aneurysm formation. The spots far from the anastomosis became stenosed within 3 months (3 tubes) when Biotubes had a wide intensity distribution, but the shape of the remaining two tubes remained unchanged for 6 months. The entire length of the bypass region was walled with an αSMA-positive cell layer, and an endothelial cell layer covered most of the lumen at 2 months. Complete endothelial laying of the luminal surface was obtained at 3 months after implantation, and a vascular wall structure similar to that of native blood vessels was formed, which was maintained even at 6 months. The stenosis was indicated to be caused by fibrin adhesion on the luminal surface, migration of repair macrophages, and granulation formation due to the overproliferation of αSMA-positive fibroblasts. We revealed the importance of Biotubes that are homogeneous, demonstrate a tensile strength > 5 N, and are implanted under appropriate antithrombotic conditions to achieve long-term patency of Biotube. Further, we clarified the Biotube regeneration process and the mechanism of stenosis. Finally, we obtained the necessary conditions for a confirmatory implant study planned shortly.

## 1. Introduction

Despite advances in endovascular therapy, bypass revascularization remains an important treatment for lower extremity arteriosclerosis obliterans, particularly in the peripheral region below the knee [1,2,3,4]. Small-diameter artificial blood vessels have problems with long-term patency, causing lower extremity amputation after revascularization, a factor with substantial life-long consequences for patients [5,6]. Autologous great saphenous vein grafts exhibit a high degree of patency and an anti-infective advantage over artificial grafts such as ePTFE [3]. However, the lack of a good quality vein graft is frequently encountered in clinical practice for patients with systemic atherosclerosis, as the great saphenous vein has often already been used for cardiac surgery or contralateral leg bypass.

Therefore, such patients have long awaited the advent of tissue-engineered vascular grafts (TEVGs) [7,8,9,10]. Among TEVGs, Biotubes are grafts developed from the patient’s body tissues based on in-body tissue architecture (iBTA) technology and are considered close to clinical application because of their biocompatibility and versatility in graft preparation [7]. Biotubes are autologous collagenous tubes obtained through a biological reaction to molds implanted subcutaneously, which is an encapsulation reaction [11]. A Biotube has been used in conjunction with artificial blood vessels to perform lower limb artery bypass surgery, causing one-year patency. We have developed several preparation molds (Biotube makers) and are currently conducting various preclinical tests for long-distance, below-the-knee lower extremity artery bypass surgery using a Biotube alone. Previously, bypass implantation of Biotubes into carotid arteries in goats resulted in complete patency in all cases for 1 month in a series of animal experiments. In this study, we performed a preliminary evaluation to optimize the Biotube preparation conditions and implantation environment to ensure reliable patency in preparation for a long-term implantation study.

## 2. Materials and Methods

### 2.1. Ethical Approval

All animals were maintained in accordance with the Guide for the Care and Use of Laboratory Animals published by the United States National Institutes of Health (NIH publication no. 85-23, revised 1996). The Oita University Animal Ethics Committee (Protocol no. 1822001) approved this research protocol.

### 2.2. Preparation of Biotube

A Biotube maker, which was a mold for making Biotubes, was prepared following a previous report [12]. The mold consists of a spiral-shaped plastic core rod with two porous stainless steel disks, having numerous pores between them. Tissue is introduced through numerous pores into the gap between the core and disks to produce a Biotube (Figure 1). Two sizes of Biotube makers were set up, each having a diameter of 86 mm and 97 mm and a disk thickness of 5.4 mm. The resulting Biotubes were 40 cm and 55 cm in length, with a 4 mm inner diameter and a 0.85 mm wall thickness. The thinness of Biotubes for clinical use while maintaining strength was considered at the mold design stage [7]. The Biotube makers were embedded subcutaneously in the abdomen of six Saanen goats (age: ≥12 months, weight: 39–70 kg; 4–6 molds per animal). Anesthesia was induced using 2 mg/kg of ketamine and maintained with 2–3% sevoflurane. The molds were harvested under general anesthesia after approximately 2 months, and Biotubes extracted from the molds were stored in 70% ethanol solution for 30 min and then preserved in 10% ethanol solution. Biotubes were then rinsed with saline solution for 10 min and prepared for the implantation procedure. A pressure leak test was performed before implantation by connecting one end of a Biotube to a syringe with a pressure gauge and applying 200 mmHg of water pressure to check for leakages. Biotube samples were placed in a portable uniaxial tensile tester (Stency, AcroEdge, Osaka, Japan), and a tensile test was performed. The rupture strength of Biotube is the load (N) at which the sample was pulled and ruptured. The sample used to determine tensile strength was 5 mm in width, and six samples were collected from each Biotube graft for evaluation. The method of these strength tests followed the previous literature [7].

### 2.3. Surgical Procedures

Anesthesia was induced as stated above in one or both carotid arteries of the same individual goat with a Biotube created under general anesthesia on the same day as the Biotube harvest. The carotid arteries were exposed and taped through two 4 cm skin incisions, spaced 7 cm apart on the goat’s neck. Heparin at 300 IU/kg was administered via the jugular vein, and additional heparin was administered as needed to maintain an activated clotting time (ACT) ≥ 300 s. The central carotid artery was cross-clamped, and end-to-side proximal anastomosis was performed using a continuous 7-0 polypropylene suture. Biotubes were passed subcutaneously to the distal skin incision, with the other side similarly anastomosed to the carotid artery distally, and the clamp was released. Native carotid arteries were treated with ligation. Postoperative antiplatelet therapy consisted of heparin (subcutaneous dalteparin at 1000 U) for 1 week and oral clopidogrel at 75 mg plus oral aspirin at 100 for 1 month. Graft patency was confirmed daily by arterial palpation from the body surface and evaluated via ultrasonography. Morphological changes over time from the anastomosis to the entire graft, blood flow in the graft lumen by color Doppler, and changes in blood flow velocity in the graft by pulsed wave Doppler were evaluated. Blood flow velocity was measured in all cases at the portion of Biotube, approximately 1 cm peripherally from the central anastomosis. Observations were performed on the day of surgery; 1, 2, 3, and 7 days postoperatively; and weekly thereafter. The target observation period was 6 months, and the grafted Biotube was removed, including the anastomotic site, in animals with open Biotubes. Biotubes, in cases where the graft lumen progressively narrowed, were removed when the lumen diameter had narrowed to ≥90% of the anterior–posterior lumen diameter because of modifications caused by thrombus formation during tissue analysis.

### 2.4. Histological Examination

The harvested Biotubes were fixed in a 4% paraformaldehyde saline solution (FUJIFILM Wako Pure Chemical Co., Osaka, Japan). Biotubes were cut open, including the anastomosis, and the lumen was macroscopically evaluated along the entire length. Biotubes were cut into strips every 1 cm, including the anastomosis, embedded in paraffin, sliced into 5 μm sections, and stained with hematoxylin and eosin. Further, anti-α-smooth muscle actin (α-SMA) mouse monoclonal antibodies (1:200; ab7817, Abcam, Cambridge, UK) were used for immunohistochemical staining. Additionally, Alexa Fluor^®^ 594 rabbit antimouse secondary antibodies were used to assess myofibroblast localization. A previous study revealed that host α-SMA positive cells, as a biological reaction after Biotube transplantation, infiltrate Biotubes during the tissue regeneration process, which may help construct new vascular wall structures and generate elastic fibers [12]. The CD31 rabbit polyclonal antibodies (1:100; ab28364, Abcam) and goat secondary antibodies to rabbit immunoglobulin G (Alexa Fluor^®^ 488) are also used to confirm vascular endothelial cell localization. DAPI (ProLong™ Gold Antifade Mountant with DAPI, Thermo Fisher Scientific, Inc., Waltham, MA, USA) was used as a nuclear counterstain.

## 3. Results

### 3.1. Preparation of Biotube

A Biotube maker with a 9.7 cm diameter was used, which is designed to produce Biotubes with a 4 mm inner diameter and a 55 cm length (Figure 1). A total of 19 Biotube makers were subcutaneously embedded (3–4 per animal) in six adult goats (39–70 kg) (Figure 2b–d). The embedding periods were set at 1 month (n = 3), 2 months (n = 3), and 5 months (n = 2) (Figure 3a). Computed tomography (CT) scan revealed no damage to the Biotube maker under the abdomen skin (Figure 2e), close contact between the Biotube makers, and the subcutaneous tissue without fluid retention around the Biotube maker (Figure 2f,g) during the embedding period. After the predetermined embedding procedure (1–6 months), all cases showed no Biotube maker inflammation, infection, or exposure (Figure 2h), except for one case where the goat scratched the wound, causing a wound infection (Figure 2i). All Biotube makers were firmly attached to the subcutaneous tissue with subcutaneous tissue infiltration in the Biotube maker through its micropores (Figure 2j). However, the Biotube maker harvest was relatively easy, as the adhesion was not strong. Biotubes were formed around a spiral rod inside all the Biotube makers (Figure 2k), except for the one case with wound infection, when the obtained Biotube makers were disassembled. Tubular Biotubes with a length of approximately 50 cm and an inner diameter of 3–4 mm were obtained as designed without any defects when the spiral rod was removed (Figure 2l). Biotubes were flexible, and their lumen remained patent without bending or occlusion even when completely bent (Figure 2m). The outer surface of Biotubes had a fine convex surface according to the Biotube maker’s micropore shape, while the luminal surface in contact with the rod was very smooth and shiny (Figure 2n).

No Biotubes leaked or ruptured even under water pressure > 200 mmHg. Figure 3b shows the results of the tensile strength test of Biotube samples cut in rings regarding Biotubes used in the later implantation experiments. The tensile strength of all Biotubes formed in all embedding periods exceeded the mean value of 2.83 ± 0.28 N of the carotid tensile strength of adult goats [7]. The mean values of the strength increased with the embedding period at 7.27 ± 1.91 N, 7.89 ± 1.81 N, and 15.94 ± 3.34 N at 1, 2, and 5 months, respectively. A weak area < 5 N and a strong area > 10 N were formed at 1 and 2 months of embedding, respectively. All areas were stronger than 10 N after 5 months of embedding. In contrast, Biotube D obtained from a goat with wound infection demonstrated weak areas.

Masson’s trichrome staining demonstrated that the Biotubes were almost entirely occupied with collagen (Figure 3). In general, they consisted of a dense luminal surface and a relatively sparse wall structure. Biotube B demonstrated discontinuous areas of insufficient collagen formation that coincided with the micropores. Biotube D exhibited internal fissures, and the collagen in Biotubes E–H was relatively densely arranged in layers.

### 3.2. Implantation of Biotube

One Biotube from each of the six goats (A–H in Figure 3a) was used to autologously bypass the carotid artery of each goat with a length of approximately 12 cm (Figure 2o,p). Goats E and G or goats F and H underwent bilateral carotid artery bypass at different periods. The Biotube could be safely grasped, and needle penetration was smooth, similar to saphenous vein grafts. There was no tearing of the Biotubes during anastomosis in all cases. After the bypass procedure, there was no bleeding or tearing from the needle hole at the anastomosis site or from the Biotube itself.

Figure 4a shows the entire process after bypass with Biotube. In Biotube A, a thrombus in the lumen of the Biotube gradually adhered to the graft immediately after implantation, and the entire graft was filled with thrombus and occluded the day after the surgery. Biotubes B and D ruptured with a small hole in the middle on postoperative days 7 and 5, respectively (Figure 4b). The other five grafts achieved bypass patency for >2 months, of which two Biotubes (E and F) remained open for 6 months. Figure 4c shows the change in ultrasonographic findings during the implantation period. All cases demonstrated no anastomotic stenosis or dilatation of Biotubes during the observation period. Stenosis occurred in Biotubes C, G, and H approximately 1 month after surgery. All stenosis occurred several centimeters away from both anastomosis parts (Figure 4b). Biotubes E and F exhibited no change in echogenic shape. Intra-graft flow velocities measured in the mid-portion of the grafts using ultrasound pulse Doppler were compared in two groups: Biotubes with stenosis (C, G, and H) and without stenosis (E and F) (Figure 4d). The flow velocity in the nonstenotic group remained constant at approximately 40–80 cm/s during the implantation period. In contrast, the flow velocity immediately after bypass in the stenotic group was high, ranging from 80 to 120 cm/s, but rapidly decreased as the stenosis progressed.

The condition of Biotube E was macroscopically observed through a skin incision 3 months after bypass. Adhesion between the Biotube and the surrounding subcutaneous tissue was mild, and the Biotube was easily dissected, similar to natural blood vessels (Figure 5a). The shape of the Biotube demonstrated no abnormal changes, such as aneurysm, over the entire length, including the anastomosis. Cutting the Biotube in the center revealed a thin white intimal layer formation without any thrombus attachment (Figure 5b).

A contrast-enhanced CT scan at 6 months after bypass revealed that although the Biotube was implanted in a curved shape to straddle the native carotid artery, its running shape significantly changed and became in line with the native artery (Figure 5c–e). No stenosis or aneurysm, including both anastomosis parts, was observed; the boundary between the graft and the natural artery was unclear, and the Biotube matched the shape of the natural blood vessel. Biotube adhesion with the surrounding area was mild even after 6 months of implantation, and its entire length could be easily removed (Figure 5f). The harvested Biotube contained very thin walls, but they did not shrink like native arteries (Figure 5g). A smooth, fresh, and shiny inner membrane layer was observed along the entire length of the Biotube when observing the lumen surface of the Biotube cut open in the longitudinal direction, and a thrombus was not observed (Figure 5h).

### 3.3. Histological Examination

Figure 6 shows the histological evaluation of the entire length of the Biotube implanted for 6 months. The wall of the implanted area was very thin along its entire length. Masson’s trichrome staining showed the presence of a collagen layer throughout the entire bypass area, similar to the Biotube before implantation, but the collagen layer corresponding to the Biotube was not apparent. The implanted area contained α-SMA-positive cells, forming a layer that extended the entire length of the graft but became thinner near the center. The layer of α-SMA-positive cells was thick and well-demarcated near the anastomosis, unlike the arrangement of the positive smooth muscle cell layer of the native carotid artery. The entire length of the luminal surface demonstrated a monolayer of CD31-positive cells, indicating that complete endothelialization had occurred. Therefore, the vascular wall structure was established within 6 months of Biotube implantation.

Tissue structural changes of Biotubes after implantation were observed over time in their central region, where blood vessel wall construction is considered to be the slowest (Figure 7). Most of the implanted Biotubes remained at 2 months. The graft wall was infiltrated with α-SMA-positive cells, but its luminal surface was only partially covered with CD31-positive endothelial cells. Biotubes existed in fragments after 3 months. An α-SMA-positive cell layer began to form inside the Biotube layer. Furthermore, a layer of CD31-positive endothelial cells completely covered the luminal surface. Therefore, the Biotube fragment remained, but a layered structure similar to the native artery wall was formed within 3 months of implantation. Biotubes almost disappeared 6 months after implantation, and the native artery wall structure remained stable, as shown above. Further, in the 6-month-implant model, Elastica van Gieson staining demonstrated the presence of elastin, thus confirming the construction of a new vascular wall.

In contrast, the locally narrowed Biotubes were also histologically examined. The area near the anastomosis where the lumen was maintained exhibited almost no fibrin adhesion on the lumen surface, and an α-SMA-positive smooth muscle cell layer covered the Biotube surface (Figure 8a). CD163-positive cells were observed, but their number was small and scattered. However, fibrin adhered to its lumen surface in the stenosis part of the Biotube, and CD163-positive repair macrophages aggregated and existed at a high density inside and directly below the fibrin layer (Figure 8b). The tissue in the wall of the stenosis appeared as granulation-like tissue, mainly composed of α-SMA-positive fibroblasts.

## 4. Discussion

We prepared small-diameter autologous Biotubes systematically, evaluated the grafts *in vivo* after carotid artery bypass surgery, and performed histopathological analysis post-harvest using six goats. Several studies involving clinical applications reported small-diameter TEVGs bypass to human coronary arteries or below-the-knee peripheral arteries mainly using heterologous autologous vessels [13]. However, the current mainstream application is development using dialysis models that do not directly affect patient life due to their poor patency results. A study on the clinical use of human acellular vessels for peripheral artery disease revealed that Lawson’s group demonstrated a 2-year secondary patency rate of 74% in femoral revascularization using a 6 mm diameter graft [9]. The result represents a major advance in the clinical use of TEVGs. In contrast, Skvorid et al. examined the factors contributing to the TEVG patency rate in their meta-analysis [14]. The TEVG experiments revealed a short graft length of 5 cm and a short observation period of 56 days on average, indicating that conditions were insufficient for direct comparison with autologous veins. Currently, many TEVGs are only in the preclinical stage using large animals.

We have focused on developing graft materials that can be used for small-diameter and long-distance bypasses as an alternative to the great saphenous vein. Our results reported the 1-year patency of small-diameter long bypass in goat carotid artery bypass surgery using Biotubes [8]. Biotubes are a product of iBTA that uses the biological reaction of encapsulation against molds and Biotube makers and do not require any scaffold materials, special environments, or techniques for cell treatments such as culture or decellularization. If the Biotube can be used as a graft with guaranteed long-term patency, iBTA will be a versatile technology that can be implemented at any general facility. However, through this experimental system, we became acutely aware that barriers must be overcome to ensure patency. The two main goals included achieving graft strength to prevent bleeding and rupture events and promoting regeneration or remodeling of the vascular wall structure as appropriate for long-term patency. These events are considered to be the reason for TEVGs not reaching clinical trials despite numerous developments. TEVGs, including Biotubes derived from living tissue, are subject to cellular responses of the host after implantation, complex biomolecular interactions, and hemodynamic conditions that are difficult to predict *in vitro*. Therefore, the indications obtained in this study toward long-term patency may provide valuable clues to solving many common issues in TEVGs.

Figure 3 shows that the strength of Biotubes was associated with the subcutaneous embedding period of the Biotube makers. The Biotube maker contained a porous outer shell and a center rod, and collagen was produced in the gap between the two parts to form the Biotube. Improvements have been made to prevent the center rod of Biotube makers from shifting to make the wall thickness of Biotubes uniform. Therefore, no Biotubes demonstrated extremely thin wall parts histologically, and the tensile strength of all Biotubes exceeded that of adult goat carotid arteries at 2.83 N [7]. However, two Biotubes failed early after implantation. Reliability after implantation could not be simply guaranteed just because of the higher strength before implantation than that of target blood vessels. Tensile strength was measured using multiple samples taken from the same graft; thus, a wide range of data naturally indicates differences in strength. Graft B failed early after implantation, indicating insufficient strength obtained during the 1-month implantation period of the mold. The systemic inflammatory response may have caused variations in the strength of the Biotube, even if the infection did not occur directly in the Biotube, in graft D, which was formed in an infected individual, even during the long embedding period of 2 months. In these cases, the Biotube structure may have immature parts in the microscopic region. In contrast, failure of the Biotube did not occur at strengths > 5 N. Therefore, an overall strength > 5 N is considered to demonstrate no weak part of the Biotube even in the microscopic region. A strength > 5 N may be generally required before implantation to prevent early failure, even in TEVGs.

During the Biotube formation process, collagen fibers accumulate from the contact surface of the core rod inside of the mold when the mold is implanted subcutaneously. Therefore, a high-density collagen layer was formed near the core rod, corresponding to the lumen of the Biotube. Our previous study revealed that the dense collagen surface prevented the penetration of blood components and the formation of blood clots when iBTA tissue was applied to the aortic valve leaflet [7,15]. Therefore, the iBTA tissue surface is considered suitable for application in cardiovascular tissues. However, graft A was occluded immediately after implantation. Goats are generally known to have high blood coagulability [16]. Further, heparin clotting time regulation in humans does not necessarily apply to goats. Therefore, the prevention of blood clots during and after surgery should be given close attention.

The next challenge toward the long-term patency of Biotube, after avoiding early postoperative thrombotic occlusion, is to properly remodel the vessel wall structure [16,17]. This study clarified Biotube reconstruction to the vascular wall structure by histologically investigating time-dependent changes in Biotubes after bypass surgery. Our previous study revealed that only vascular endothelial progenitor cells adhered to the luminal surface, except near both anastomoses, 1 month after Biotube implantation. Endothelial cell adhesion was only localized 2 months after Biotube implantation. Complete endothelialization along the entire length of the luminal surface was achieved 3 months after implantation and was maintained and stabilized after 6 months. In contrast, the wall tissue of the Biotube itself became fragmented 2 months after implantation and was replaced by a layer containing α-SMA-positive cells along its entire length. A-SMA-positive cells covered the lumen and outer surfaces sandwiching the Biotube tissue. Biotube tissue had almost disappeared 6 months after implantation, and the replacement of the Biotube with blood vessel wall tissue was completed. Generally, endothelialization occurs both by continuous luminal migration from the native artery to the implanted graft and by derivation from bone marrow cell-derived endothelial progenitor cells supplied by the bloodstream [17,18]. Further, endothelialization using both methods occurred in Biotubes. Therefore, endothelialization can be expected along the entire length even with longer Biotubes. As endothelial cells mature, signaling pathways cause the appropriate expression of smooth muscle cells, resulting in stable remodeling. The grafts, which obtained patency for 6 months, were expected to remain patent for a longer period.

Three grafts (C, G, and H) exhibited wall thickening approximately 1 month after the bypass procedure, followed by stenosis (Figure 3). However, both anastomoses demonstrated no abnormal shape change at the graft parts in all three cases, and the graft stenosis occurred at a short distance from both anastomoses. Through a series of experiments, we believe that the fragmentation of these Biotubes is a process of scaffold induction of autologous cells and regeneration of new vascular structures [8]. Generally, intimal hyperplasia (IH), which reduces the vascular lumen due to excessive vascular smooth muscle cell proliferation and extracellular matrix deposition, is an important factor in the failure to achieve the long-term patency of TEVGs [17]. The presence of vascular endothelial cells is crucial to maintaining graft patency during the appropriate remodeling process of the vascular wall structure. Endothelial cells secrete antithrombotic factors, such as nitric oxide and tissue plasminogen activator, and play an important role in maintaining the homeostasis of blood vessel walls [17,19]. In contrast, when endothelial cells are damaged, macrophages adhere to the vascular lumen and secrete chemical mediators, such as TNF-α, MMP, and IFNγ, which cause the attachment of platelets and leukocytes, thus resulting in thrombosis. These mediators induce the differentiation of transplanted hematopoietic stem cells into myofibroblasts, which further promotes collagen deposition and causes IH, thus resulting in graft dysfunction [17]. These events were observed only as early as 6 weeks after implantation [19], coinciding with the onset of graft stenosis or reduced flow in this study (Figure 4c,d).

As mentioned above, external factors unrelated to the physical properties of Biotubes were considered because the stenosis of Biotubes occurred locally and was delayed. Simulations of saphenous vein graft stenosis after lower extremity bypass surgery using computational fluid dynamics [20,21,22] and a discussion of the angle and shape of the anastomosis [23] have been reported, and the phenomenon of IH in the vascular graft was discussed from a fluid dynamics perspective. Flow geometrical changes, stagnation, and turbulence are risks in the local graft area because bypass surgery is essentially an artificial blood flow design that differs from the line of a native artery. Wall shear stress (WSS), which is a hydrodynamic index, or time-averaged WSS, which excludes the effect of heart rate variability on WSS, has been associated with IH and discussed from a molecular biological perspective [20,22,24]. Furthermore, thrombus formation or IH development, from a molecular biological perspective, can be induced at sites of blood stagnation, where endothelial cell homeostasis is not maintained and macrophage engraftment is enhanced [17]. Haruguchi et al. [23] have explained that the hemodynamic difference between arterial bypass grafts (ABGs) and arteriovenous grafts (AVGs) is due to the difference in their blood flow rates; the blood flow rate of AVGs is 5 to 10 times higher than that of ABGs, and the IH that occurs in AVGs is caused by endothelial cell damage. This may also be related to the higher flow velocity in the stenosis group during the early phase after implantation (Figure 4d); however, it is difficult to explain IH in a unified manner because various factors are involved. Our countermeasure is to avoid stenosis by drawing the anastomotic angle gently with special consideration to the anastomotic shape to avoid low WSS; however, understanding blood flow stagnation through actual post-bypass blood flow analysis is a future subject.

Herein, we investigated the regeneration process of blood vessel walls after Biotube implantation and discussed methods to obtain long-term patency for clinical trials in an *in vivo* performance test of an autologous Biotube induced by iBTA technology as a preclinical test using large animals. Biotubes require a period of subcutaneous formation, which limits their emergency use. However, Biotube medicine may be versatile enough to be accepted by any facility because special scaffold materials or cell treatment processes are not required when preparing Biotubes. In contrast, the experiment was conducted on goats with hypercoagulability, thus setting special conditions for goats, such as an ACT > 300 s, which is different from that for humans, and animal experiments demonstrated limitations despite using large animals. Additionally, this study was at a preliminary stage; thus, the small overall number of cases and the lack of patency cases indicated limitations. Evaluation of elastin was thought to be essential for histologically proving the replacement of Biotubes with vascular tissue; however, this evaluation was only possible in some samples. Currently, we are conducting a confirmation test in a 6-month implant model and are pursuing long-term patency for over 1 year. Additionally, we plan to clarify the regeneration mechanism of the Biotubes through ongoing confirmation tests.

## 5. Conclusions

We revealed the importance of Biotubes that are homogeneous, demonstrate a tensile strength > 5 N, and are implanted under appropriate antithrombotic conditions to achieve long-term potency of Biotubes. Further, we clarified the Biotube regeneration process and mechanism of stenosis. Finally, we obtained the necessary conditions for a confirmatory implant study planned shortly.

## Figures and Tables

**Figure 1 bioengineering-11-00203-f001:**
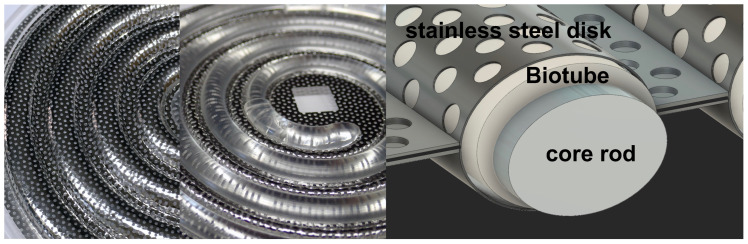
Photograph and schema of the mold. The plastic core rod is sandwiched between two stainless steel disks. Once implanted subcutaneously, the gap between the core and disks is filled with fibroblasts and the collagen they produce through the numerous holes formed in the disks over a period of approximately two months.

**Figure 2 bioengineering-11-00203-f002:**
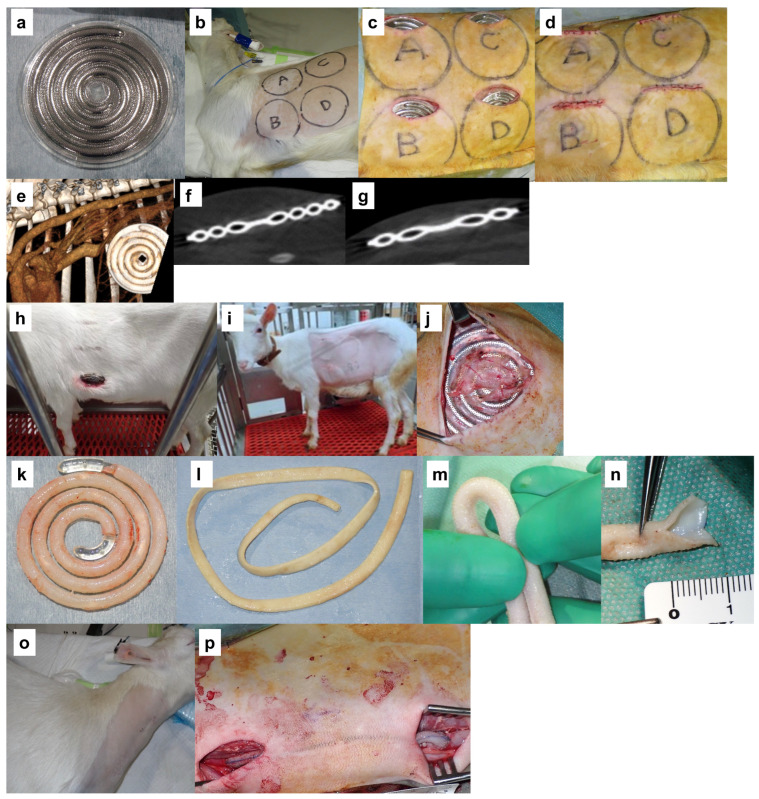
Biotube preparation and bypass surgery. (**a**) Porous stainless steel molds, Biotube maker. (**b**–**d**) Before, during, and after embedding the Biotube makers subcutaneously in a goat. Frontal (**e**) and cross-sectional (**f**,**g**) computed tomography (CT) images during Biotube maker implantation. (**h**) The only Biotube maker to have had a wound infection caused by a goat scratch. (**i**) In all goats, the skin was clean after embedding without redness, swelling, or exposure of the Biotube makers. (**j**) During the harvest, the Biotube maker was covered with subcutaneous tissue. (**k**) Biotube was formed around the spiral rod inside the mold. (**l**) Biotube was obtained by removing it from the spiral rod. (**m**) Biotube does not kink under 100 mmHg, and (**n**) the luminal surface was smooth. (**o**) Pre- and (**p**) post-bypass surgery.

**Figure 3 bioengineering-11-00203-f003:**
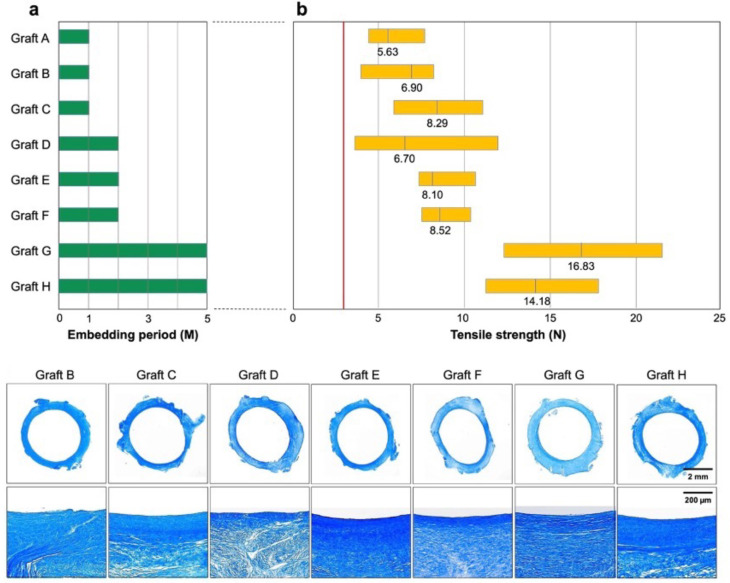
(**a**) Duration of subcutaneous Biotube maker implantation of eight Biotube grafts (Biotube A–H) used for bypass and (**b**) distribution of their respective tensile strength. The red line indicates the tensile strength of normal goat carotid arteries. Masson’s trichrome staining demonstrates that Biotubes were almost entirely occupied with collagen.

**Figure 4 bioengineering-11-00203-f004:**
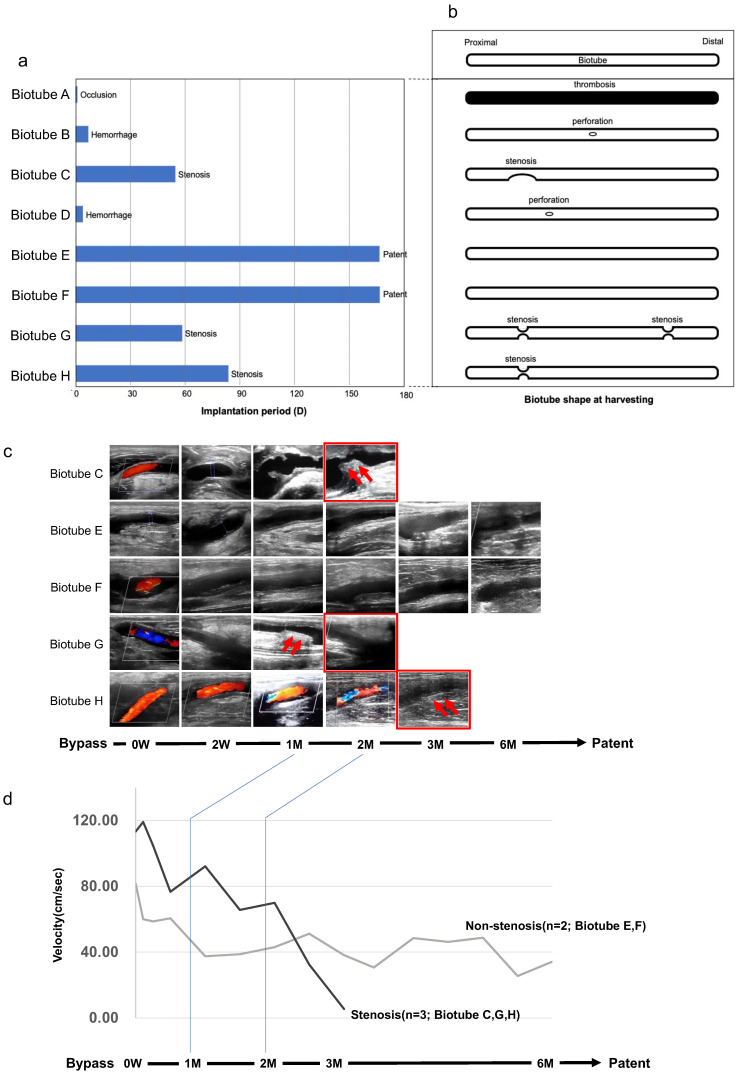
(**a**) Implantation period and implantation status after bypass using Biotube. (**b**) Schematic diagram of Biotube shape change. (**c**) Echocardiographic findings after bypass. Arrows indicate the findings of intimal thickening on echographic images, which are observed around 2 months after bypass (Biotube C, G, H). (**d**) The time course of flow velocity in Biotube after implantation. The stenosis group tended to have higher flow velocity in the immediate postoperative period. A rapid decrease in flow velocity was observed 1–2 months after bypass.

**Figure 5 bioengineering-11-00203-f005:**
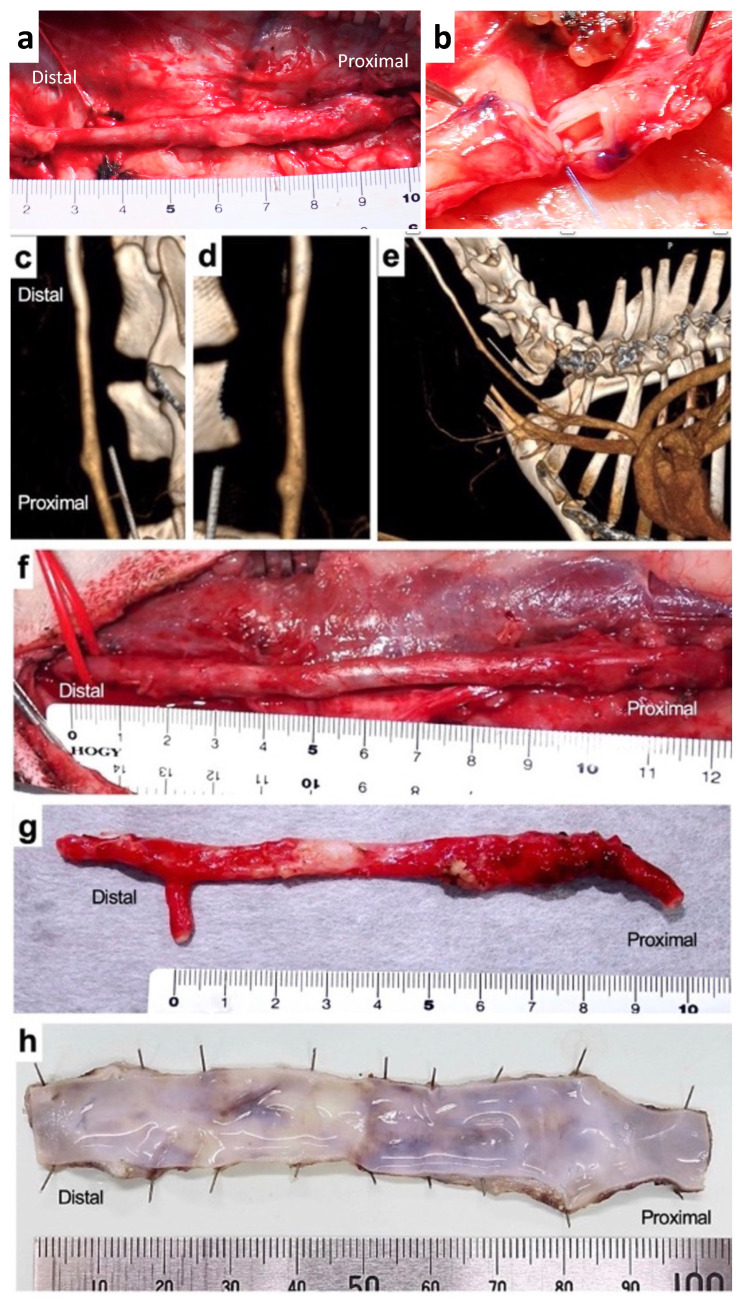
(**a**) Biotube bypass graft dissected from the surrounding subcutaneous tissue 3 months after bypass. (**b**) Cross-section of the bypass graft from the center of photo (**a**). (**c**–**e**) Contrast-enhanced CT scan 6 months after bypass showing the linear Biotube graft, similar to the native carotid artery, with no stenosis or aneurysm formation. (**f**) Biotube bypass graft dissected from the surrounding subcutaneous tissue 6 months after bypass. (**g**) Macroscopic view and (**h**) the luminal surface of the harvested bypass graft.

**Figure 6 bioengineering-11-00203-f006:**
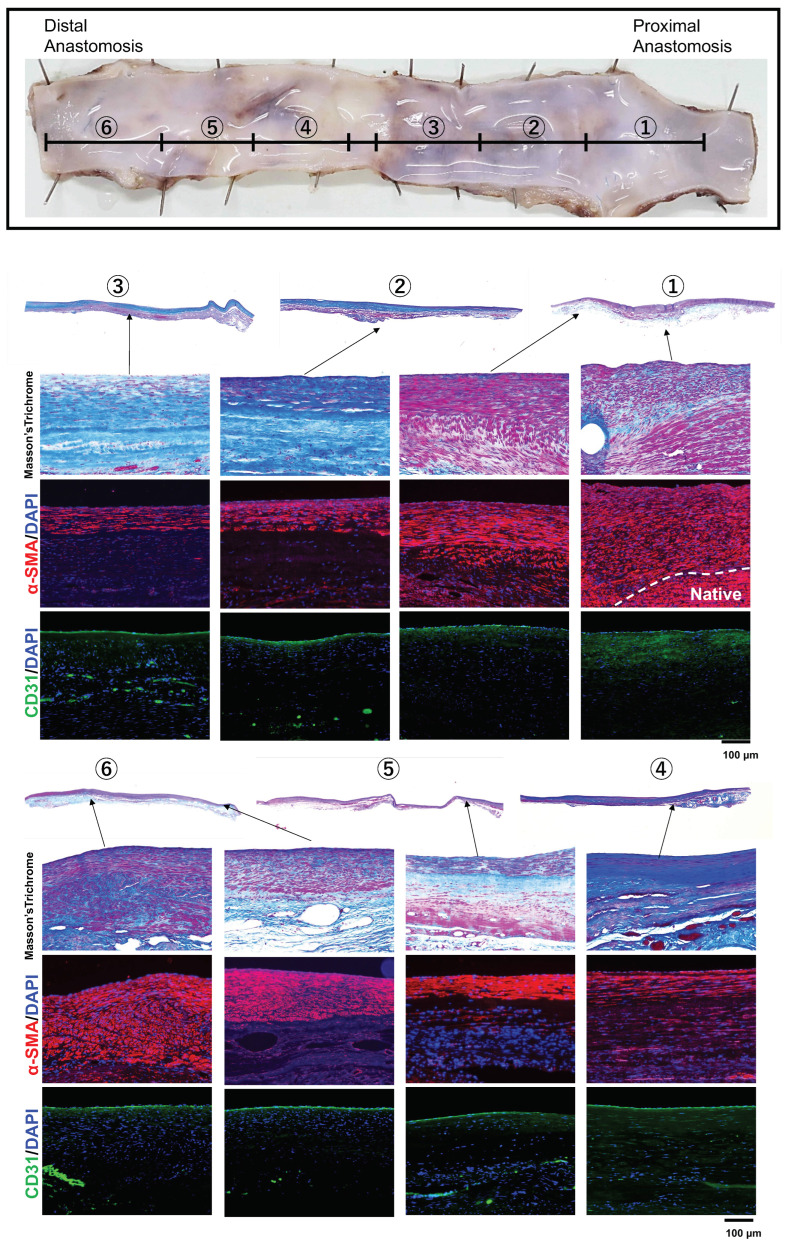
Histological observation of the entire length of Biotube graft implanted for 6 months. Masson’s trichrome staining illustrating the presence of collagen throughout the entire length of Biotube, which was different from the collagen in the Biotube. α-SMA-positive cells are thicker near the anastomosis and thinner toward the center of the Biotube, and the arrangement of α-SMA-positive cells near the anastomosis is different from that of native α-SMA-positive smooth muscle cells. CD31-positive endothelial cells are found throughout the entire length of the Biotube. These findings indicate the establishment of blood vessel wall architecture.

**Figure 7 bioengineering-11-00203-f007:**
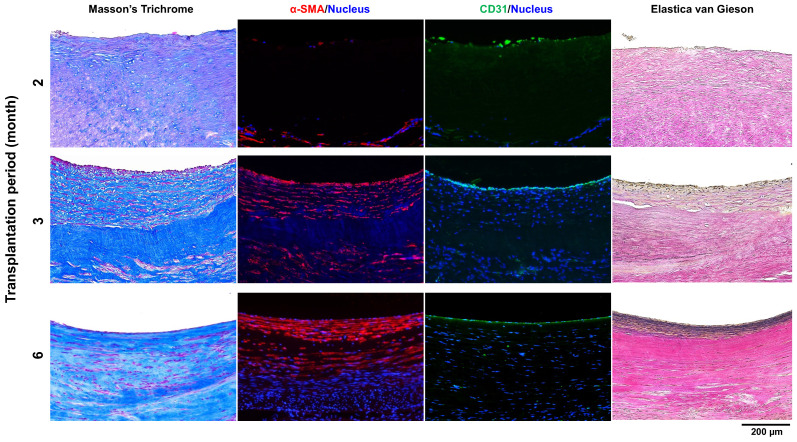
Histological photos of the central portion of Biotube grafts, which take time to reconstruct the blood vessel wall, were observed over time at 2, 3, and 6 months. Most of the Biotube structure remained, and α-SMA-positive cells began to infiltrate, limited to partially covering endothelial cells, at 2 months. A layer of α-SMA-positive neoplastic cells had formed, and endothelial cells were covering the entire graft at 3 months. At 6 months, the presence of elastin fibers was observed using Elastica van Gieson staining. The original Biotube structure was gone, and the blood vessel wall was reconstructed as a vascular wall at 6 months.

**Figure 8 bioengineering-11-00203-f008:**
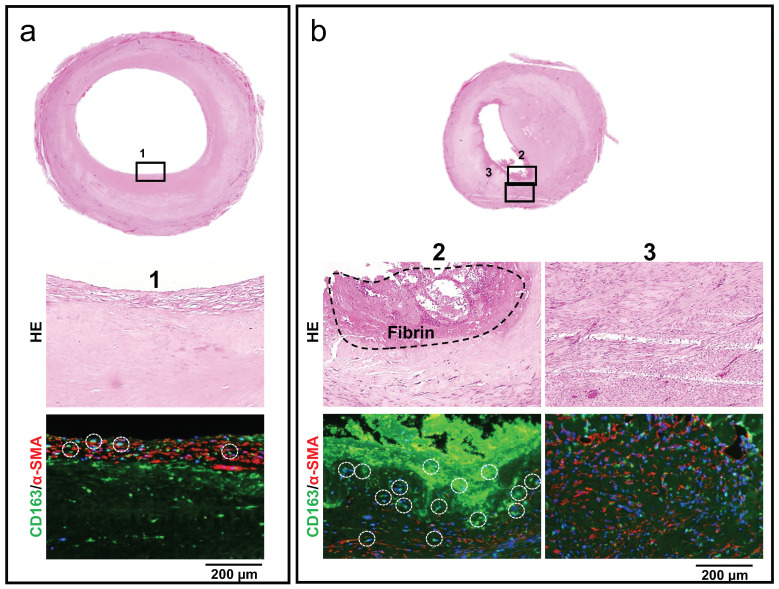
Comparison of stenotic and nonstenotic areas of partially stenotic Biotube grafts. (**a**) The lumen surface is smooth in the area without stenosis, and the Biotube is covered with α-SMA-positive cells. CD163-positive repair macrophages (dashed circles) are present but not numerous (region 1). (**b**) The stenosis demonstrated the adherence of fibrin clot to the luminal surface and a high concentration of CD163-positive repaired macrophages just below the fibrin layer (region 2). Additionally, the vessel wall demonstrated granulation-like tissue growth by α-SMA-positive fibroblasts (region 3).

## Data Availability

The data are not publicly available due to their containing information that could compromise the privacy of research participants.

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
