# Peer review of "Carotid Artery Bypass Surgery of In-Body Tissue Architecture-Induced Small-Diameter Biotube in a Goat Model: A Pilot Study"

_bioengineering, 2024, doi:10.3390/bioengineering11030203_

Round 1

Reviewer 1 Report

Comments and Suggestions for Authors

Bioengineering 2823738

Carotid artery bypass surgery of in-body tissue architecture-in-2 duced small-diameter Biotube in a goat model: A pilot study

In this work authors investigate the performances of in vivo generated TEVG (biotubes). The manufacturing process is presented and the performances of these devices are studied in a goat model in carotid position. Results bring out that some device remain fully functional and patent after 6 months implantation

The paper is well written and presents an interesting TE approach. It provides results that are can be useful for future developments in the field of TE. However, some points need to be completed or changed:

-“…The mold consists of a spiral-shaped plastic core rod with two porous stainless steel disks…” not clear what the purpose of the disks is , a schema would help

-“…with a 4-mm inner diameter and a 0.85-mm wall…” how do these specificatiosn correspond to ? why 0.85 thickness…?

-“…without any defects when the spiral rod was removed…” what about adhesion issues ? could the finally observed fragmentation be related to local degradation induced at the rod removing step ?

-“…the Biotubes were almost entirely occupied with collagen…” where does the collagen come from ? Foreign Body Reaction ? this should be better discussed in the discussion section

-“…Biotube. Graft A was occluded 198 by thrombus the day after surgery…” not clear what the explanation for that is

-Figure 3 is not clear and should be reconsidered: too many snapshots, not clear what these represent etc…

-regarding Fig 3d: why does the velocity go down over time in the non-stenosed vessels….?

-“…Biotube existed in fragments after 3 months…” why ? initial collagen is not supposed to get resorbed with time….

-“…Biotube almost disappeared 6 months after…” what is the mechanism involved ? actually the Biotube is already collagen, again,  why should the material disappear ?

-“…the anastomosis where the lumen was maintained exhibited almost no fibrin adhesion…” why ? stress is expected to be higher in that zone, restenosis is expected to appear etc….what the mechanisms involved based on literature data ….?

-“…The two main goals included achieving graft strength to prevent bleeding and rupture events and promoting regeneration or remodeling….” Why remodeling ? the mock vessel is already functional ?

-“…Reliability after implantation could not be simply guaranteed just because of the higher strength before implantation …” why ? where does the loss of strength come from ? Literature data should be provided here to explain the mechanisms induced…like collagen loss of strength …based on what ? etc…

-“…Biotube structure may have immature parts in the microscopic region…” has this been analyzed or is that just an assumption ? in the 2nd case, justifications should be provided (literature data ?)

-“…graft A was occluded immediately…” mechanism ? blood or restenosis ?

-“…Biotube itself became fragmented 2 months…” again why ?

-“…Additionally, the timing of graft stenosis coincided with that 394 of flow decline…” not clear what is meant

Author Response

Thank you very much for taking the time to review this manuscript. I'll add detailed answers to each comment below.

Sincerely,

Reviewer1

-“…The mold consists of a spiral-shaped plastic core rod with two porous stainless steel disks…” not clear what the purpose of the disks is , a schema would help

A new schema has been created (Fig1).We cited our past paper to explain the preparation principle of Biotubes. Biomaterials 2018, 185: 232-239.

Briefly, The mold consists of a spiral plastic core sandwiched between two porous stainless steel plates, and fibroblasts and the collagen fibers they produce penetrate in the gap between the core and plates, forming a Biotube.

-“…with a 4-mm inner diameter and a 0.85-mm wall…” how do these specificatiosn correspond to ? why 0.85 thickness…?

In a previous report [7], the wall thickness was 0.85 mm to be as thin and strong as possible for clinical use. Explanation added: (L85-86)

-“…without any defects when the spiral rod was removed…” what about adhesion issues ? could the finally observed fragmentation be related to local degradation induced at the rod removing step ?

There was little adhesion between the rod and Biotube. The Biotube removal operation was extremely smooth. In addition, since we have confirmed that there was no leakage under water pressure at 200 mmHg, we believe that there is no damage caused by the removal operation.

-“…the Biotubes were almost entirely occupied with collagen…” where does the collagen come from ? Foreign Body Reaction ? this should be better discussed in the discussion section

There was a lack of explanation for the encapsulation reaction seen as a biological reaction to the template. Explanation added. (L61-62)

-“…Biotube. Graft A was occluded by thrombus the day after surgery…” not clear what the explanation for that is

Detailed description added,( L187-190)

-Figure 3 is not clear and should be reconsidered: too many snapshots, not clear what these represent etc…

I have carefully selected snapshots and organized the figures. (Fig4-c,d)

-regarding Fig 3d: why does the velocity go down over time in the non-stenosed vessels….?

The graph shows that the non-stenotic group sustained moderate flow velocities for six months. The graph has been modified to make it clearer.(Fig4-d)

-“…Biotube existed in fragments after 3 months…” why ? initial collagen is not supposed to get resorbed with time….

The process of regeneration of new blood vessels by replacing the Biotube scaffold with neoplastic cells is additionally described in the "Discussion" section.(L343-345)

-“…Biotube almost disappeared 6 months after…” what is the mechanism involved ? actually the Biotube is already collagen, again,  why should the material disappear ?

The process of regeneration of new blood vessels by replacing the Biotube scaffold with neoplastic cells is additionally described in the "Discussion" section.(L341-343)

-“…the anastomosis where the lumen was maintained exhibited almost no fibrin adhesion…” why ? stress is expected to be higher in that zone, restenosis is expected to appear etc….what the mechanisms involved based on literature data ….?

In general, endothelial cells may invade continuously from the native vessel (L330-333), and we believe that endothelial cells had already formed at the anastomosis and did not occlude it. It was clear that the stenosis of the anastomosis was not the cause of this occlusion.

-“…The two main goals included achieving graft strength to prevent bleeding and rupture events and promoting regeneration or remodeling….” Why remodeling ? the mock vessel is already functional ?

Biotubes are replaced by autologous cells and act as scaffolds for regeneration [8,15]. In fact, in this 6-month model, the original Biotube structure had almost disappeared. This is what we described as regeneration.

-“…Reliability after implantation could not be simply guaranteed just because of the higher strength before implantation …” why ? where does the loss of strength come from ? Literature data should be provided here to explain the mechanisms induced…like collagen loss of strength …based on what ? etc…

Both Graft G and H, which were stronger in the rupture test, failed to achieve six-month patency. Not only were both grafts stronger in the rupture test, but there were also variations in strength (Fig. 3-b). We found that even if the break test assures strength, the patency is not always as good as that of autologous vessels. Long-term patency is affected not only by collagen strength, but also by endothelial cell function and hemodynamic factors such as WSS, as discussed in the discussion. I do not yet have data to fully explain all of these events.

-“…Biotube structure may have immature parts in the microscopic region…” has this been analyzed or is that just an assumption ? in the 2nd case, justifications should be provided (literature data ?)

We were not able to present a photo of the Biotube, but the wall of the excised Biotube was thinning in some areas, where it had ruptured. The guess is that the thinning was caused by a problem with the individual goat, and that previous infection may have caused insufficient Biotube formation in the skin ulcer on goat’s back.

-“…graft A was occluded immediately…” mechanism ? blood or restenosis ?

Graft A believes that the thrombus occlusion was caused by factors such as insufficient heparin use and the time required for the surgery in the early stages of this study.

-“…Biotube itself became fragmented 2 months…” again why ?

The process of regeneration of new blood vessels by replacement of the Biotube scaffold with neoplastic cells is additionally described in the "Discussion" section. We have already seen resorption of the Biotube in the early postoperative period.

-“…Additionally, the timing of graft stenosis coincided with that of flow decline…” not clear what is meant

I expressed that the timing of 6 weeks, when IH generally occurs, was very close to the timing of the decrease in flow velocity in the stenosis group in our experiment. Expression was changed.

Reviewer 2 Report

Comments and Suggestions for Authors

This study made autologous tubular tissues named Biotubes, and demonstrated implantation performance in a goat model. The results of this pilot study can be expected for future clinical applications. I have the following comments for the authors.

1. Overall

This study does not show post-operation breaking strength. The authors explain that the tissue in the Biotube tissue disappear and is replaced after transplantation. The graft failure occurred even though the strength was higher than that of a native vessel wall, so isn't the strength pre-transplantation indirect data?

2. Overall

I think the arterial walls also contain elastin fibers. Histological evaluation of elastin is also important in demonstrating the replacement of Biotubes tissue into the vessel wall.

3. Introduction

Please consider describing a concise preparation principle of Biotubes for the reader’s understanding.

4. L92-95

Please describe the sample size. If the unit is load (N), the rupture strength depends on the sample size.

5. L172-174

In Fig. 2, the arrangement of layers in the circumferential direction cannot be clearly determined.

6. Figure 3d

Please describe data variation.

7. L344-345

I don't think it is possible to discuss whole TEVG because the strength is based on the results obtained from a pilot study of Biotubes. If this sentence is a general fact mentioned in TEVG, please provide references.

8. L368-370

After 6 months of implantation, have the Biotubes been completely replaced by arterial wall with a laminated structure of elastin? Please consider showing the results or changing the explanation.

9. L429-430

I think the mechanism of stenosis is not clear, and this study has provided a hypothesis for the cause of stenosis.

Author Response

Thank you very much for taking the time to review this manuscript. I'll add detailed answers to each comment below.

Sincerely,

Reviewer2

  1. Overall This study does not show post-operation breaking strength. The authors explain that the tissue in the Biotube tissue disappear and is replaced after transplantation. The graft failure occurred even though the strength was higher than that of a native vessel wall, so isn't the strength pre-transplantation indirect data?

All of the obtained Biotubes had higher tensile strength than native vessels. However, Biotubes with a pressure under 5N caused rupture after implantation. It is thought that Biotubes with a strength under 5N had locally weak parts. Therefore, as pointed out by the reviewer, the measured strength did not reflect the entire area of Biotube and was considered to be only indirect data. However, we believe that if the strength is over 5N, it can be determined that there are no locally weak areas, and it is useful for evaluation before implantation.

  1. Overall I think the arterial walls also contain elastin fibers. Histological evaluation of elastin is also important in demonstrating the replacement of Biotubes tissue into the vessel wall.

As the reviewer points out, evaluation of elastin is important to demonstrate the replacement of Biotubes tissue into the vessel wall. This study is a preliminary study to determine the conditions for a non-clinical study. Furthermore, since the N number is small, we can only estimate the replacement of Biotubes tissue to the blood vessel wall. Based on this research, we are currently conducting a non-clinical study for long-term implantation. We performed Elastica van Gieson staining in Fig7 to try to prove the presence of elastin. We are investigating in detail the mechanism of replacement of Biotubes tissue into the vessel walls, including observation of elastin.

  1. Introduction Please consider describing a concise preparation principle of Biotubes for the reader’s understanding.

A new schema has been created (Fig1).We cited our past paper to explain the preparation principle of Biotubes. Biomaterials 2018, 185: 232-239.

Briefly, The mold consists of a spiral plastic core sandwiched between two porous stainless steel plates, and fibroblasts and the collagen fibers they produce penetrate in the gap between the core and plates, forming a Biotube.

  1. L92-95 Please describe the sample size. If the unit is load (N), the rupture strength depends on the sample size.

Samples were taken from 6 locations for all grafts L100-101.The size of the sample used for tensile strength was 5 mm in width.

  1. L172-174 In Fig. 2, the arrangement of layers in the circumferential direction cannot be clearly determined.

The notation has been revised (L174-178).

  1. Figure 3d Please describe data variation.

Changed the graph to show that graft stenosis correlates with decreased flow velocity (FIG.4-d)

  1. L344-345 I don't think it is possible to discuss whole TEVG because the strength is based on the results obtained from a pilot study of Biotubes. If this sentence is a general fact mentioned in TEVG, please provide references.

I quoted from the description of the importance of these chemical mediator trends in TEVG in general[20].

  1. L368-370 After 6 months of implantation, have the Biotubes been completely replaced by arterial wall with a laminated structure of elastin? Please consider showing the results or changing the explanation.

As mentioned above, the evaluation of elastin is currently underway with the intention of describing it in detail in the next report.

  1. L429-430 I think the mechanism of stenosis is not clear, and this study has provided a hypothesis for the cause of stenosis.

We believe that the mechanism of IH cannot be explained in a unitary manner. However, the fact that reference [26]concludes that a high Blood flow rate is a factor in the development of IH could support the fact that IH progression was marked in the grafts with high flow velocity in Figure 4. Additional description. In addition, by keeping in mind the anastomosis angle to rise gently, stenosis can be clearly avoided.(L370-380)

Round 2

Reviewer 1 Report

Comments and Suggestions for Authors The R1 version is ok for me now.

Comments on the Quality of English Language fine